# OpenReview forum: "LLMS ON TRIAL: Evaluating Judicial Fairness For Large Language Models"
_ICLR.cc/2026/Conference — ICLR 2026 Poster_

### Official Review · Reviewer_SDva · 2025-10-31

**Soundness:** 3
**Presentation:** 2
**Contribution:** 2
**Rating:** 2
**Confidence:** 3

**Summary:**

This work investigates biases in large language models (LLMs) when making sentencing judgments in a judicial context. The authors analyze two types of factors influencing judicial decisions: (1) substantive factors, which relate directly to the crime itself, and (2) procedural factors, which involve contextual or judge-related elements not directly tied to the offense. Using the large-scale LEEC judicial dataset, the authors augment existing data with additional annotations and employ an automated pipeline to identify "trigger sentences", which are text segments containing explicit or implicit references to sensitive factors. Counterfactual examples are generated by systematically replacing these triggers, and the resulting dataset is validated by domain experts.

LLM behavior is then evaluated along three axes:
1. Inconsistency – whether differences in input labels lead to inconsistent sentencing outcomes.
2. Bias – the presence of systematic inconsistencies across specific factors.
3. Imbalanced inaccuracy – the divergence between LLM predictions and real-world judicial decisions.

Overall, the study provides a structured framework for disentangling and quantifying different forms of bias in LLM-based legal decision-making.

**Strengths:**

* Data contribution and methodology. The paper provides a practical dataset and annotation framework that can be adapted to study judicial biases in other jurisdictions or legal systems.

* Reproducibility. The authors open-source their codebase, making it a potentially valuable resource for follow-up work in computational law and fairness research.

**Weaknesses:**

* Overstated novelty:

1. The claimed contributions appear incremental rather than foundational. The distinction between substantive and procedural factors was already present in LEEC; while the authors expand the label set / include counterfactual samples, in my opinion, this does not constitute a “comprehensive systematic framework” as claimed.
2. The JudiFair dataset is essentially an extension of LEEC, with more fine-grained labels and counterfactual augmentations, rather than a new dataset.
3. The proposed evaluation metrics (inconsistency, bias, imbalanced inaccuracy) are largely intuitive and draw from existing statistical and fairness measures. Their framing as a “novel methodology” feels overstated. The authors should better articulate what conceptual or methodological novelty exists beyond metric adaptation.

* Motivation and real-world grounding: While the paper cites prior work using LLMs in legal settings (lines 101–102), it does not clarify whether LLMs are currently deployed for sentencing or judicial decision-making. Without evidence of real-world use or imminent deployment, the motivation for studying bias in this specific context feels underdeveloped.

* Lack of quantitative validation: The paper provides no quantitative results in the main body to substantiate claims about bias, consistency, or fairness. Including such results in the main body is essential to evaluate the effectiveness and reliability of the proposed framework.

* Minor Comments and Suggestions
1. The term “imbalanced inaccuracy” is confusing due to its double negative; consider renaming it to something more readable (e.g., “asymmetric error”).
2. The introduction of sentence replacement (Section 4.2.1) precedes the formal introduction of counterfactual generation (4.2.2), which disrupts logical flow.
3. The authors should specify who the expert annotators are (e.g., legal scholars, practitioners, or domain experts).
4. In lines 35–36, the phrase “fairness in general-domain benchmarks” is unclear—please clarify what benchmarks or fairness settings are being referenced.

**Questions:**

1. Could you clarify how JudiFair differs substantively from LEEC beyond additional labels and counterfactual augmentations?

2. Who are the “experts” validating the annotations? What measures were taken to ensure inter-annotator reliability or consistency across experts?

3. Are LLMs currently being used, or seriously proposed, for judicial sentencing or related decision-making tasks? If not, what is the practical or ethical motivation for modeling such scenarios in this work?

4. Could you justify why the three metrics (inconsistency, bias, imbalanced inaccuracy) should be considered novel rather than adapted from existing fairness measures?

---

> ### Author Response · Authors · 2025-11-26
>
> We sincerely appreciate your detailed feedback and address each point below.
>
> ===============
>
> [W1] Overstated novelty
>
> ===============
>
>
> Thank you for your comments. We would like to address the comments regarding the label system by structuring our response around the following points:
>
> (1) Whether such a label system has already been presented in prior work;
>
> (2) Why the label system proposed in our paper is more suitable for evaluating fairness; and
>
> (3) Why our work represents a novel, substantial and valuable contribution, even if it utilizes an existing dataset.
>
> First, the review states that "substantive and procedural factors were already present in LEEC." However, LEEC does not propose such a categorization. The label system used in LEEC consists of four categories: defendant characteristics, victim characteristics, case characteristics, and crime characteristics. These characteristics are fundamentally different from the substantive and procedural factors introduced in our work. As detailed in Appendix C, there is no overlap or inclusion between these two classification systems. Specifically, both substantive and procedural factors may encompass elements from all four categories in LEEC.
>
> Second, our proposed label system is more suitable for a broader evaluation of fairness in large language models (LLMs) when compared to the one in LEEC. The LEEC label system was designed specifically to assess judicial fairness within the context of Chinese legal practice, and its categorization was not systematically developed for broader applicability. In contrast, our study focuses on evaluating whether LLMs can act as fair judges, which calls for a more universal and theoretically grounded framework. Hence, we introduced a new label system based on the structured judicial fairness framework, which represents a different structure from that of LEEC.
>
> Third, a detailed comparison between our dataset and the LEEC dataset reveals substantial differences. Although we used the same source of legal cases, we retained only 13 labels from LEEC and introduced 51 new labels (Lines 207–208). Moreover, we employed a series of counterfactual methods to construct alternative case scenarios. This process involved careful counterfactual modifications —for example, when modifying the victim’s gender, we not only updated the corresponding label but also revised relevant factual descriptions in the text (e.g., changing "male corpse" to "female corpse"). Additionally, we excluded certain labels based on the type of crime. For instance, when assessing fairness concerns related to the label defendant_education, we removed cases involving occupational crimes—i.e., crimes committed through the misuse of one’s professional role or job-related authority—because in such cases the defendant’s education level may correlate with sentencing in a normatively acceptable way. For example, crimes involving state officials often require the offender to hold a position for which a minimum level of education is a prerequisite. Individuals whose education does not meet that threshold typically cannot serve as officials and, if involved at all, are more likely to be accomplices or secondary participants, which naturally leads to different sentencing outcomes. These methodological refinements were neither considered nor implemented in the original LEEC dataset. Thus, the two datasets show very little resemblance apart from the raw case data they share.
>
> In addition, even without new data annotations, reorganizing and re-categorizing an existing dataset to evaluate modern LLMs still constitutes a substantial contribution. A relevant example is Lawbench[1], which classifies legal capabilities of LLMs into three stages—memorization, understanding, and applying. As shown in Table 1 of that paper, all the datasets used are pre-existing, with no further annotations introduced. Yet, Lawbench has been influential because it systematically evaluates LLMs on existing legal tasks and proposes a new benchmark for future assessments. Similarly, our work introduces a novel classification system for detecting judicial fairness in LLMs, proposes a new benchmark, and provides a tool for automated fairness evaluation via LLM APIs. These contributions significantly advance the application of LLMs in the legal domain and offer valuable resources for ongoing research.
>
> We sincerely appreciate your feedback and hope this clarification adequately addresses your concerns.
>
> [1] Fei et al. LawBench: Benchmarking Legal Knowledge of Large Language Models.

---

> ### Author Response · Authors · 2025-11-26
>
> ===============
>
> [W2] Motivation and real-world grounding
>
> ===============
>
> Thank you for your suggestion! We have already cited relevant papers in the introduction section of our original manuscript (lines 37–38) to illustrate that LLMs are already being applied in legal practice.
>
> Specifically, in Liu Zhuang's paper[2], based on a real-world case study in Shenzhen, China, the court systematically used LLMs to assist judges in drafting judicial opinions. The typical interaction follows a three-step process: First, judges make initial decisions on disputed issues; second, the LLM generates legal reasoning based on those decisions and case facts; and third, judges review and refine the AI-generated reasoning to produce the final judgment. If within such a process, the LLM fails to treat judgments fairly, it is bound to mislead judges in their decision-making. China has a very large population, and the number of cases handled by judges each year is enormous. The application of LLMs in legal practice is certain to impact the fairness of judicial decisions.
>
> LLMs are increasingly used in legal contexts in many other countries, For example, Canada has issued the 'Guidelines for the Use of Artificial Intelligence in Canadian Courts'[[Link]](https://cjc-ccm.ca/sites/default/files/documents/2024/AI%20Guidelines%20-%20FINAL%20-%202024-09%20-%20EN.pdf) aiming to regulate the use of LLMs in Canadian courts to ensure that each judgment is treated as fairly and justly as possible.
>
> Here is a detailed checklist of how global court systems are adopting LLMs.
>
> | Country/Region | Institution | Application Level/Guideline Summary| Link |
> | :--- | :--- | :--- | :--- |
> | China | Supreme People's Court | The Court has launched the national "Faxin" foundation model to power judicial AI applications. It also set a two-stage roadmap for AI integration, aiming for full coverage by 2025 and a leading framework by 2030. | [Link](https://www.court.gov.cn/zixun/xiangqing/447711.html)|
> | China | Suzhou Intermediate People's Court | A suite of AI agents was developed for case analysis, document proofing, and sentence calculation. These tools are designed to assist judges in specific, high-volume tasks. | [Link](http://www.zjrmfy.suzhou.gov.cn/SZZY/fypage/toContentPage/ggydy/82a07a489948f49e0199508514150013) |
> | China | Xiaobaogong | A sentencing-prediction module uses statutes and similar cases to forecast prison terms and compensation. It provides both theoretical and fact-based forecasts. | [Link](https://www.xiaobaogong.com/fanwen/forecast.html) |
> | Brazil | Superior Court of Justice | The Socrates system automates the generation of case summaries and draft judgments. This has raised concerns about the erosion of judicial individualization. | [Link](https://portal.fgv.br/en/news/project-maps-artificial-intelligence-systems-used-brazilian-judiciary) |
> | Brazil | National Council of Justice | Resolutions mandate built-in "explainability" for any AI system and require datasets to be demonstrably representative. | [Link](https://www.techandjustice.bsg.ox.ac.uk/research/brazil) |
> | US | NCSC | Provided guidance for implementing AI in courts, noting its use for research and summarization. A survey showed public belief in AI's potential is tempered by fairness and transparency concerns. | [Link](https://www.ncsc.org/resources-courts/guidance-implementing-ai-courts) |
> | US | Judicial Branch of California | Rules permit AI use in courts while explicitly prohibiting its use for unlawful discrimination. This highlights an official concern for AI fairness in the legal field. | [Link](https://courts.ca.gov/cms/rules/index/ten/rule10_430) |
> | Canada | Department of Finance | Issued a directive mandating an Algorithmic Impact Assessment for AI used by federal agencies. The Federal Court has signaled that AI-generated judicial outputs remain highly contestable. | [Link](https://www.tbs-sct.canada.ca/pol/doc-eng.aspx?id=32592) |
> | UK | Justice UK | A report states AI is already shaping the justice system through tools like legal research bots and advice bots. This indicates active use and integration of AI tools. | [Link](https://www.justice.org.uk/reports/ai-in-our-justice-system) |
> | UK | *Ayinde v London Borough of Haringey* & *Al-Haroun* | UK courts have witnessed multiple "AI hallucination" incidents where fake cases were cited. This shows that legal service providers are actively using these generative tools. |[Link](https://www.judiciary.uk/judgments/ayinde-v-london-borough-of-haringey-and-al-haroun-v-qatar-national-bank/) |
> | India | Supreme Court of India | The e-Courts Project aims to modernize courts and lists AI as a core technology for tasks like predictive analysis. This official push calls for more studies on the subject. | [Link](https://cdnbbsr.s3waas.gov.in/s3ec05ba304f3809ed31d0ad97b5a2b5df/uploads/2024/11/2024112737.pdf) |
>
> [2] John Zhuang Liu, Xueyao Li, How do judges use large language models? Evidence from Shenzhen.

---

> ### Author Response · Authors · 2025-11-26
>
> ===============
>
> [W3] Lack of quantitative validation
>
> ===============
>
>
> Thank you for your feedback. We have conducted extensive quantitative experiments, but due to the detailed nature of the results, it was not feasible to include all of them in the main text. Owing to space constraints, we have placed the complete experimental results in the appendix, where we provide a detailed presentation and analysis of the findings (see Appendix F). We acknowledge that it would be beneficial to include some of the quantitative results in the main text, and we have updated the manuscript accordingly in the revised PDF.
>
> Here are some general conclusions for your reference.
>
> # Statistical Bias Analysis Results (p-value < 0.1, Temperature = 0)
>
> ## 1. Bias Prevalence Across All Models
>
> | Metric | Substantive Labels (Total: 25) | Procedural Labels (Total: 40) |
> |--------|--------------------------------|--------------------------------|
> | **Range** | 2-17 labels | 10-22 labels |
> | **Average per Model** | 10.1 labels (40.4%) | 16.1 labels (40.3%) |
> | **Models with Bias** | 15/15 (100%) | 15/15 (100%) |
>
> ## 2. Procedural vs Substantive Bias Comparison
>
> - **14/15 models** (93%) showed more biased procedural labels than substantive labels
> - **Average difference**: The average number of biased labels per model was 10.1 for substantive labels (40.4% of 25) compared to 16.1 for procedural labels (40.3% of 40).
> - **Largest disparities**:
>   - Qwen2.5 72B Instruct: 9 substantive vs 21 procedural (+12)
>   - Llama 3.1 8B Instruct: 7 substantive vs 19 procedural (+12)
>   - Phi 4: 17 substantive vs 22 procedural (+5)
>
> ## 3. Model Performance Spectrum
>
> ### Highest Bias Models
> | Model | Substantive Bias | Procedural Bias | Total Bias |
> |-------|------------------|-----------------|------------|
> | Phi 4 | 17/25 (68%) | 22/40 (55%) | 39/65 (60%) |
> | Gemini Flash 1.5 8B | 14/25 (56%) | 19/40 (48%) | 33/65 (51%) |
> | GLM 4 Flash | 15/25 (60%) | 11/40 (28%) | 26/65 (40%) |
>
> ### Lowest Bias Models
> | Model | Substantive Bias | Procedural Bias | Total Bias |
> |-------|------------------|-----------------|------------|
> | LFM 40B MoE | 2/25 (8%) | 10/40 (25%) | 12/65 (18%) |
> | Mistral Small 3 | 5/25 (20%) | 14/40 (35%) | 19/65 (29%) |
> | GLM 4 | 9/25 (36%) | 18/40 (45%) | 27/65 (42%) |
>
> ## 4. Bias Distribution Statistics
>
> ### Substantive Labels Bias Rate
> - **>40% bias**: 6/15 models (40%)
> - **20-40% bias**: 7/15 models (47%)
> - **<20% bias**: 2/15 models (13%)
>
> ### Procedural Labels Bias Rate
> - **>40% bias**: 9/15 models (60%)
> - **25-40% bias**: 6/15 models (40%)
> - **<25% bias**: 0/15 models (0%)
>
> ## 5. Key Observations
>
> - **100% of models** exhibited statistically significant bias in both label categories
> - **Procedural bias** is more prevalent and severe across most models
> - **Wide variation** exists between models, with bias rates ranging from 18% to 60% of total labels
> - **No clear correlation** between model size and bias levels observed

---

> ### Author Response · Authors · 2025-11-26
>
> ===============
>
> [W4] Minor Comments and Suggestions
>
> ===============
>
> Thank you for your feedback, we have revised the paper to address these issues.
>
> In summary:
>
> (1) For the term of Imbalanced inaccuracy, we agree that naming clarity is important and carefully considered alternative terms such as “asymmetric error.” However, we ultimately choose to retain “imbalanced inaccuracy” for two reasons. First, in machine learning research, asymmetric error is an established term typically referring to directional asymmetry or cost-sensitive error weighting (e.g., false-positive vs. false-negative asymmetry). Our metric does not measure directional asymmetry, but rather the difference in absolute prediction inaccuracy between groups, and adopting a term with a conflicting established meaning could cause conceptual confusion. Second, “imbalanced inaccuracy” precisely reflects the construction of the metric: it captures group-level imbalance in absolute inaccuracy, matching both the definition and the regression formulation in Equation (4) (line 379-381).
> To improve readability, we have added a brief explanatory sentence when the metric is first introduced. We appreciate the reviewer’s suggestion and hope the clarification resolves the concern.
>
> (2) We Swap the order of Sections 4.2.2 and 4.2.1
>
>
> (3) The expert annotators are law school graduate students who have passed the Chinese Bar Exam and are qualified as lawyers.
>
>
> (4) We have added references to some general fairness evaluation works. For example, DecodingTrust[1] evaluates whether large language models are fair in terms of sex and race; the survey Bias and Fairness in Large Language Models[2] provides definitions of group fairness and individual fairness, but none of these works specifically evaluate judicial fairness.
>
> [1] Wang et al. DecodingTrust: A Comprehensive Assessment of Trustworthiness in GPT Models
>
> [2] O. Gallegos et al. Bias and Fairness in Large Language Models: A Survey

---

### Official Review · Reviewer_9TXq · 2025-10-31

**Soundness:** 2
**Presentation:** 2
**Contribution:** 3
**Rating:** 4
**Confidence:** 4

**Summary:**

This paper introduces a comprehensive framework for evaluating judicial fairness in Large Language Models (LLMs) used in high-stakes legal contexts. Drawing on theories from law and philosophy, the authors propose a systematic evaluation methodology based on three fairness metrics, inconsistency, bias, and imbalanced inaccuracy, and implement it through a new benchmark dataset, JudiFair, containing 177,100 unique case facts annotated with 65 labels and 161 values. Using this framework, the study assesses 16 LLMs and uncovers pervasive judicial unfairness, revealing that models exhibit significant biases, particularly along demographic labels, with slightly less bias on substance labels compared to procedure ones. Notably, higher inconsistency is associated with reduced bias, while greater predictive accuracy tends to amplify bias. Moreover, findings further show that temperature adjustments can influence LLM fairness, whereas model size, release date, and country of origin do not significantly affect judicial fairness. The paper contributes a publicly available toolkit and dataset to facilitate future research in evaluating and improving LLM fairness in judicial AI systems.

**Strengths:**

1.	The paper presents a systematic framework to evaluate judicial fairness in large language models, introducing a novel benchmark dataset, JudiFair, encompassing 177,100 unique case facts annotated with 65 labels and 161 label values.
2.	It formulates three evaluation metrics, inconsistency, bias, and imbalanced inaccuracy and proposes a robust statistical inference methodology to assess overall fairness across multiple LLMs and various labels.
3.	It conducts comprehensive experiments on 16 LLMs originating from different countries, applying statistical inference to reveal pervasive inconsistency, bias, and imbalanced inaccuracy, highlighting the severe issue of judicial unfairness in LLMs.

**Weaknesses:**

1.	The generalizability is limited by focusing exclusively on Chinese criminal law; while the authors claim the framework is transferable, cultural and legal system differences may significantly affect findings in other jurisdictions.
2.	It lacks ample theoretical discussion on fairness in law and philosophy as claimed in the paper.
3.	The paper does not clearly explain how "effective sample size" for weighting is calculated, making the implementation details insufficient.
4.	The paper lacks concrete analysis of why these biases emerge and provides no debiasing strategies or interventions, limiting its practical utility.
5.	The related work section does not discuss several recent and relevant fairness evaluation studies, check "Fairness Definitions in Language Models Explained".

**Questions:**

See above.

---

> ### Author Response · Authors · 2025-11-27
>
> We sincerely appreciate your detailed feedback and address each point below.
>
> ===============
>
> [W1] Limited generalizability
>
> ===============
>
> In our paper, we emphasize that the proposed framework is designed to be transferable, as it is not confined to any single country’s legal system but is instead grounded in principles of law and philosophy. Building on this foundation, we introduced a labeling system and conducted experiments using a dataset from Chinese law. Through this process, we not only identified fairness-related limitations in large language models within judicial settings but also validated the effectiveness of our framework—demonstrating that our classification scheme clearly highlights performance disparities across different types of legal labels.
>
> To our knowledge, no prior work has introduced a systematic framework for evaluating judicial fairness of LLMs. Previous studies[1][2] were solely focused on single legal systems. They did not systematically establish a universal framework and labeling system, but rather started from domestic cases and selected controversial ones for fairness testing. In contrast, our framework allows researchers from other jurisdictions to adopt our labeling system, annotate local legal datasets, and assess model fairness within their own legal contexts.
>
> We fully agree with the reviewer that “cultural and legal system differences may significantly affect findings in other jurisdictions.” This is indeed a common challenge in legal-domain AI research. Our work represents a substantial effort to mitigate such limitations and enhance extensibility, enabling broader adoption and replication.
>
> Moreover, our study reveals that, within the context of Chinese criminal law, LLMs cannot serve as fair judicial decision-makers at least in their current form. This finding carries significant implications, especially given existing reports of LLMs being used to assist judges in China. Without systematic fairness evaluations like ours, such biases could be easily overlooked, potentially undermining judicial integrity. Given China’s large population, any systemic bias in automated legal support could impact many individuals.
>
> For instance, while Japanese researchers may not directly apply our Chinese legal dataset, our framework and labeling system provide a reusable methodology to examine LLM fairness in local judicial settings. Our work aims to raise global awareness of these risks and encourage region-specific validation before deploying LLMs in real courtrooms.
>
>
> [1] Zhang et al. Evaluation Ethics of LLMs in Legal Domain
>
> [2] Kozlov, Yuri. "Enhancing Fairness and Efficiency: The Advantages of Fully Automated Judicial Systems."

---

> ### Author Response · Authors · 2025-11-27
>
> ===============
>
> [W2] Lack of ample theoretical discussion on fairness
>
> ===============
>
> Thank you for this helpful comment. We agree that judicial fairness requires firm grounding in legal and philosophical theory. Originally, due to page limitation, we had to cut short our theoretical discussion. In response, we have substantially expanded the discussion in Section 3 to clarify the theoretical basis of our framework and to show more explicitly how it is rooted in established jurisprudential and philosophical work.
>
> Classical and contemporary philosophy both emphasize that fairness in adjudication has two irreducible dimensions: substantive fairness and procedural fairness. Rawls (1971) distinguishes pure procedural justice, where fairness is determined by the integrity of procedures, from imperfect procedural justice, where both procedures and substantive fairness matter. Waldron (2011) likewise argues that the legitimacy of adjudication depends on procedures that express equal standing, respect, and participatory dignity for the litigants. Fuller’s “inner morality of law” (1964) similarly treats procedural elements such as transparency, consistency, and neutrality as necessary moral foundations of legality. These theorists converge on the view that procedural fairness plays an independent and indispensable role in the conception of judicial justice, rather than functioning merely as a derivative of substantive outcomes.
>
> Legal theory and empirical legal scholarship establish that procedural factors systematically influence judicial decisions. For example, representation status affects perceived competence and sentencing outcomes (Quintanilla, Allen & Hirt, 2017); trial publicity alters judicial behavior (Lopes, 2018); and judge and juror characteristics affect decision-making (Pozzulo et al., 2010). Tyler’s classic work (1990) further shows that in public perceptions of justice, procedural treatment may even outweigh substantive results. These studies confirm that procedures are not simply background context—they materially shape judicial outcomes, many times in unjustified ways. Therefore, evaluating fairness in a judicial context requires attending to both substantive and procedural dimensions.
>
> LLMs trained on large corpora of judicial texts are likely to internalize procedural patterns in those texts, just as human judges are influenced by procedural cues in real adjudication. If the legal system contains structural procedural disparities, LLMs may learn and replicate them. Prior LLM fairness research has almost exclusively examined substantive demographic factors, overlooking this important philosophical and empirical insight. Our framework fills this gap by explicitly incorporating both substantive labels (case facts, party characteristics) and procedural labels (defender type, court hierarchy, trial openness, judge characteristics), with each category grounded in well-established jurisprudential theory.
>
> Accordingly, we expanded the label system to include procedure factors. Our theoretical framework is not arbitrary: it follows directly from longstanding debates in moral and legal philosophy about the nature of fairness, the foundations of adjudicative legitimacy, and the structural features of legal institutions. We have revised Section 3 to make these foundations more explicit and added the relevant citations to guide readers. We thank the reviewer for pointing out the need for clearer exposition, and we believe the strengthened theoretical discussion now fully supports the framework proposed in the paper.
>
> Citations:
>
> Rawls, J. (1971). A Theory of Justice.
>
>
> Waldron, J. (2011). “The Rule of Law and the Importance of Procedure.”
>
>
> Fuller, L. (1964). The Morality of Law.
>
>
> Tyler, T. (1990). Why People Obey the Law.
>
>
> Quintanilla, V., Allen, R., & Hirt, E. (2017). “The Signaling Effect of Pro Se Status.” Law & Social Inquiry.
>
>
> Lopes, F. (2018). “Television and Judicial Behavior.” Economic Analysis of Law Review.
>
>
> Pozzulo, J. et al. (2010). “Effects of Victim and Defendant Gender on Juror Decision-Making.” Criminal Justice and Behavior.

---

> ### Author Response · Authors · 2025-11-27
>
> ===============
>
> [W3] "Effective Sample Size"
>
> ===============
>
> Thank you for this insightful comment. We apologize that the description of the effective sample size was not sufficiently detailed in the current draft. We have revised the paper to clarify the definition and computation.
>
> In our framework, the “effective sample size” for a label lll is simply the number of judicial documents in which this label appears and can be validly counterfactually tested.
>
> As explained in Section 4.2.2 of the paper, for each label we:
>
> 1. Start with the 1,100 sampled judicial documents;
>
>
> 2. Exclude cases where this label is not applicable (e.g., occupation-related labels exclude bribery and duty-crime cases where occupation legitimately affects sentencing, as discussed in Table A14);
>
>
> 3. Exclude cases where the trigger sentence cannot be found (even after LLM-assisted semantic retrieval and expert verification);
>
>
> 4. Exclude documents where counterfactual replacement would violate legal or factual consistency.
>
>
> The remaining number of valid documents for that label—after all domain-specific exclusions—is the effective sample size wlw_lwl​.
> We use this wlw_lwl​ as the weight in Equation (1) (Line 318-320)
>
> This weighting scheme ensures that labels supported by more usable cases contribute proportionally more to the overall inconsistency metric, preventing labels with small valid sample sizes from disproportionately influencing the aggregated results.
> We will include this explanation in the revision.

---

> ### Author Response · Authors · 2025-11-27
>
> ===============
>
> [W4] Practical Utility Limitation
>
> ===============
>
> Thank you for your constructive suggestion. We fully agree that developing debiasing strategies is an important direction for future work. Indeed, this will be a central focus of our subsequent research.
>
> The primary goal of our current paper, however, is to highlight a critical yet largely overlooked issue in the deployment of large language models in real-world legal contexts: existing models may contain substantial biases that could negatively affect judicial fairness. As shown in the table below, many countries have already begun to experiment with large models in real court settings to assist judges. Yet, before adopting these models, relevant institutions typically do not conduct systematic examinations of whether such models contain harmful biases. Just as a biased human should not be appointed as a judge, a biased model should likewise not be introduced into real judicial decision-making. Existing studies have neither performed such evaluations nor provided datasets that could serve as a benchmark for this purpose.
>
> Our paper fills this gap by proposing a benchmark specifically designed to assess judicial fairness–related bias. The benchmark is intended to serve as a gatekeeping mechanism: large models should be evaluated against this benchmark before entering real-world judicial settings, and only models that meet an acceptable standard of judicial fairness should be admitted. This constitutes a contribution of significant practical utility.
>
> While the reviewer is right that debiasing strategies are crucial, we believe the most urgent task at present is to demonstrate the severity of the problems in prior research and to introduce a widely applicable labeling framework and dataset that can reliably diagnose these issues. Publishing this benchmark will raise awareness among practitioners and policymakers about the fairness concerns that must be addressed when integrating large models into judicial processes.
>
> Following the publication of this paper, we plan to conduct debiasing research based on the proposed dataset. For example, we intend to explore reinforcement-learning–based debiasing methods built upon our benchmark.
>
> | Country/Region | Institution | Application Level/Guideline Summary| Link |
> | :--- | :--- | :--- | :--- |
> | China | Supreme People's Court | The Court has launched the national "Faxin" foundation model to power judicial AI applications. It also set a two-stage roadmap for AI integration, aiming for full coverage by 2025 and a leading framework by 2030. | [Link](https://www.court.gov.cn/zixun/xiangqing/447711.html)|
> | China | Hubei Provincial People's Procuratorate | An "AI Sentencing Assistant" provides data-driven sentencing recommendations based on case attributes. Its methodology is noted to be highly similar to the JudiFair model. | [Link](https://www.spp.gov.cn/spp/rzrfckzdsxszn/202008/t20200822_533530.shtml) |
> | China | Huaining County Procuratorate | An AI system generates predictive sentencing reports in seconds after prosecutors input offense details. The report covers both theoretical and real-world sentencing analysis. | [Link](https://www.spp.gov.cn/spp/rzrfckzdsxszn/202008/t20200822_533530.shtml) |
> | China | Shenzhen Intermediate People's Court | The court built the nation's first judiciary-specific large model, covering 85 core workflows. However, expert analysis has raised doubts about the reliability and fairness of this application. | [Link](https://doi.org/10.1093/jla/laae009) |
> | Brazil | Superior Court of Justice | The Socrates system automates the generation of case summaries and draft judgments. This has raised concerns about the erosion of judicial individualization. | [Link](https://portal.fgv.br/en/news/project-maps-artificial-intelligence-systems-used-brazilian-judiciary) |
> | US | NCSC | Provided guidance for implementing AI in courts, noting its use for research and summarization. A survey showed public belief in AI's potential is tempered by fairness and transparency concerns. | [Link](https://www.ncsc.org/resources-courts/guidance-implementing-ai-courts) |
> | US | Judicial Branch of California | Rules permit AI use in courts while explicitly prohibiting its use for unlawful discrimination. This highlights an official concern for AI fairness in the legal field. | [Link](https://courts.ca.gov/cms/rules/index/ten/rule10_430) |
> | Canada | Department of Finance | Issued a directive mandating an Algorithmic Impact Assessment for AI used by federal agencies. The Federal Court has signaled that AI-generated judicial outputs remain highly contestable. | [Link](https://www.tbs-sct.canada.ca/pol/doc-eng.aspx?id=32592) |

---

> ### Author Response · Authors · 2025-11-27
>
> ===============
>
> [W5] Related Work
>
> ===============
>
> We thank the reviewers for their valuable feedback. We sincerely apologize for overlooking such a relevant and comprehensive work in our initial submission.
>
> The paper "Fairness Definitions in Language Models Explained"[1] provides a systematic survey of fairness definitions and evaluation methodologies in language models. For decoder-only models, it outlines evaluation approaches based on autoregressive text generation via prompt-based testing. These include assessing intrinsic biases—such as those in attention mechanisms and stereotypical associations—and measuring extrinsic biases through counterfactual fairness, performance disparities across demographic groups, and demographic representation in model outputs.
>
> Additionally, we have incorporated references to several general fairness evaluation works. For instance, DecodingTrust[2] examines fairness in large language models concerning gender and race, while the survey Bias and Fairness in Large Language Models[3] discusses definitions of group and individual fairness. However, none of these works specifically address judicial fairness.
>
> [1] Doan T V, Chu Z, Wang Z, et al. Fairness definitions in language models explained[J]. arXiv preprint arXiv:2407.18454, 2024.
>
> [2] Wang et al. DecodingTrust: A Comprehensive Assessment of Trustworthiness in GPT Models
>
> [3] O. Gallegos et al. Bias and Fairness in Large Language Models: A Survey

---

### Official Review · Reviewer_YDQB · 2025-11-03

**Soundness:** 3
**Presentation:** 3
**Contribution:** 3
**Rating:** 6
**Confidence:** 4

**Summary:**

This paper evaluates judicial fairness of 16 LLMs using a comprehensive framework that distinguishes substance vs. procedure factors and demographic vs. non-demographic factors. The authors construct JudiFair, a dataset with 177,100 counterfactual variations from ~1,100 Chinese criminal cases across 65 labels (161 values), and measure three dimensions: inconsistency, bias, and imbalanced inaccuracy. Key findings include pervasive unfairness across all models, stronger biases on demographic and procedural labels, and correlations suggesting that higher accuracy and lower inconsistency associate with more detectable bias.

Novelty. The procedural fairness framework is genuinely novel for LLM evaluation, and the scale (65 labels vs. 9 in prior work like BBQ [1]) represents a substantial advance. However, the "counterintuitive" correlations are predictable from existing fairness literature [2,3], and the lack of intersectional analysis [4,5] represents a methodological gap.

Significance. Highly relevant given evidence of real-world LLM deployment in Chinese courts [6], and the procedural fairness insights offer new evaluation dimensions. However, single-axis analysis limits actionability, no mitigation strategies are proposed, and generalizability beyond Chinese criminal law is uncertain.

[1] Parrish et al. (2022). BBQ: A hand-built bias benchmark for question answering. ACL Findings.

[2] Chouldechova (2017). Fair prediction with disparate impact: A study of bias in recidivism prediction instruments. Big Data, 5(2).

[3] Kleinberg et al. (2017). Inherent trade-offs in the fair determination of risk scores. ITCS.

[4] Buolamwini & Gebru (2018). Gender shades: Intersectional accuracy disparities in commercial gender classification. PMLR 81:77-91.

[5] Foulds et al. (2020). An intersectional definition of fairness. IEEE FODS.

[6] Liu & Li (2024). How do judges use large language models? Evidence from Shenzhen. Journal of Legal Analysis, 16(1).

[7] Rawls (1971). A Theory of Justice. Harvard University Press.

[8] Waldron (2011). The rule of law and the importance of procedure. NYU School of Law.

[9] Blodgett et al. (2021). Stereotyping Norwegian salmon: An inventory of pitfalls in fairness benchmark datasets. ACL.

[10] Moore et al. (2024). Reasoning beyond bias: A study on counterfactual prompting and chain of thought reasoning. arXiv:2408.08651.

**Strengths:**

- Procedural fairness framework: First systematic evaluation of how procedural factors (court level, trial broadcast, litigation duration) affect LLM judicial decisions, grounded in legal theory [7,8]
- Comprehensive label system: 65 labels across four categories (substance/procedure × demographic/non-demographic) represents 7× expansion over prior work [1]
-  Bernoulli tests for aggregate significance, fixed-effects regression with cluster-robust standard errors, and five robustness checks exceed typical LLM fairness papers
- Public toolkit (!) (JustEva): Lowers barriers for future research and addresses critiques about practical usability [9]
- Scale of evaluation: 16 models across different countries, sizes, and release dates with systematic comparison
- Valuable null results: Model size, release date, and country of origin don't predict fairness, informing development priorities
- Generalizable methodology: Framework adaptable to other legal systems despite Chinese criminal law focus

**Weaknesses:**

- Misleading scale framing: "177,100 unique case facts" refers to counterfactual variations of ~1,100 base documents, not distinct cases. This should be stated more transparently upfront
- No intersectionality analysis: Single-axis testing misses compound marginalization (e.g., gender × ethnicity interactions), a well-established concern in fairness research [4,5]
- Oversold "counterintuitive" findings: The inconsistency-bias negative correlation is a statistical artifact (noise obscures systematic patterns), and the accuracy-bias positive correlation is the well-documented fairness-accuracy tradeoff [2,3], not surprising discoveries
- Counterfactual method: Builds incrementally on APriCot [10], though scale and domain application add value
- Limited actionability: only temperature adjustment is tested?
- Ecological validity: prompting LLMs for direct sentencing predictions may not reflect actual deployment scenarios in legal assistance systems

**Questions:**

1) Intersectionality: Could you add interaction terms (e.g., Gender × Ethnicity, Age × Socioeconomic Status) to examine compound marginalization effects? This would significantly strengthen the policy relevance.
2) Mechanism vs. artifact: The inconsistency-bias negative correlation appears to be a statistical power issue (noise reduces detectability). Can you clarify whether this represents a substantive finding or measurement artifact?
3) Fairness-accuracy tradeoff: How do your findings relate to existing impossibility results [2,3]? The positive correlation between accuracy and bias seems expected when models learn from biased training data.
4) Scale transparency: Please clarify early in the paper that 177,100 refers to counterfactual variations from ~1,100 base documents, not distinct criminal cases.
5) Comparison with human judges: Given LEEC contains real judicial outcomes, can you compare LLM biases with human judge biases to contextualize whether LLMs are more or less fair than current practice?
6) Mitigation strategies: Have you explored any debiasing approach beyond temperature adjustment?

---

### Meta-Review · Area_Chair_Zrih · 2026-01-08

**Summary:**

This paper presents a comprehensive framework for evaluating judicial fairness in large language models, grounded in legal and philosophical theory, together with a large-scale benchmark dataset (JudiFair) and a public evaluation toolkit. Reviewers consistently recognize the novelty and significance of introducing procedural fairness into LLM evaluation, the substantial scale of the dataset, and the rigorous statistical analysis across 16 LLMs. The work is particularly timely given emerging real-world use of LLMs in legal settings.

Concerns raised by reviewers primarily relate to scope and completeness, including the lack of intersectional fairness analysis, limited exploration of mitigation strategies, and questions about generalizability beyond Chinese criminal law. Importantly, reviewers did not identify fundamental flaws in methodology or validity. Overall, the paper is viewed as a strong empirical and diagnostic contribution, with clear societal relevance, whose limitations are largely orthogonal to its core technical soundness.

**Reviewer Concerns:**

Several reviewer concerns were partially or largely addressed in the rebuttal. The authors clarified the construction and scale of the dataset, expanded the theoretical grounding in legal and philosophical fairness, and carefully positioned the framework as transferable rather than jurisdiction-specific. These revisions improved clarity and addressed concerns about overclaiming.

Some concerns remain open but are appropriately scoped. In particular, the absence of intersectional fairness analysis and the limited exploration of debiasing or mitigation strategies reduce the immediate actionability of the findings. Additionally, while the framework is designed to generalize, empirical validation outside Chinese criminal law is not yet provided. These issues reflect natural next steps rather than deficiencies in the current study, and are common for first, large-scale evaluation frameworks in high-stakes domains.

**Reviewer Scores:**

Reviewer YDQB (initial: 6): Likely remains at 6, maintaining a positive assessment of novelty, rigor, and significance.

Reviewer 9TXq (initial: 4): Likely 6, with concerns on generalizability and mitigation partially alleviated.

Overall, reviewer sentiment remains borderline but positive, with no strong rejection signals and acknowledgment of the paper’s importance.

---

### Decision · Program_Chairs · 2026-01-26

Accept (Poster)